# RHOA Therapeutic Targeting in Hematological Cancers

**DOI:** 10.3390/cells12030433

**Published:** 2023-01-28

**Authors:** Juliana Carvalho Santos, Núria Profitós-Pelejà, Salvador Sánchez-Vinces, Gaël Roué

**Affiliations:** 1Lymphoma Translational Group, Josep Carreras Leukaemia Research Institute (IJC), 08916 Badalona, Spain; 2Laboratory of Immunopharmacology and Molecular Biology, Sao Francisco University Medical School, Braganca Paulista 01246-100, São Paulo, Brazil

**Keywords:** small GTPase, RHOA, hematological cancers, lymphoma, tumor suppressor gene, oncogene, therapeutic target

## Abstract

Primarily identified as an important regulator of cytoskeletal dynamics, the small GTPase Ras homolog gene family member A (RHOA) has been implicated in the transduction of signals regulating a broad range of cellular functions such as cell survival, migration, adhesion and proliferation. Deregulated activity of RHOA has been linked to the growth, progression and metastasis of various cancer types. Recent cancer genome-wide sequencing studies have unveiled both *RHOA* gain and loss-of-function mutations in primary leukemia/lymphoma, suggesting that this GTPase may exert tumor-promoting or tumor-suppressive functions depending on the cellular context. Based on these observations, RHOA signaling represents an attractive therapeutic target for the development of selective anticancer strategies. In this review, we will summarize the molecular mechanisms underlying RHOA GTPase functions in immune regulation and in the development of hematological neoplasms and will discuss the current strategies aimed at modulating RHOA functions in these diseases.

## 1. Introduction

RHO family proteins belonging to the Ras superfamily are highly conserved proteins found in nearly all eukaryotes. In humans, this family comprises 20 members classified in eight subgroups according to their sequence and function [1]. Among them, the Ras homolog gene family member A (RHOA) is the most studied and best characterized protein. RHOA regulates several signal transduction pathways, such as the structure of cytoskeletons, gene expression, vesicle trafficking, cell proliferation, morphogenesis, cell migration, and polarity of cells [2]. It has been shown that RHO GTPases drive cell migration through their ability to promote protrusive force associated with the extension of lamellipodium, the assembly of contractile actin and myosin filaments (stress fibers) and the cell adhesions to the extracellular matrix required to allow the body and rear of the cell to follow the movement [3].

Most Rho GTPases switch between an active GTP-bound and an inactive GDP-bound conformation. The cycling between these states is regulated by three sets of proteins, guanine nucleotide exchange factors (GEFs), GTPase activating proteins (GAPs) and guanine nucleotide dissociation inhibitors (GDIs). GEFs promote the exchange of GDP for GTP necessary for the activation of GTPase, while GAPs catalyze the GTP hydrolysis of the GTPase, thereby inactivating it. GDIs inhibit the activity of RHO proteins by holding GTPases in an inactive state in the cytosol and by preventing the action of GEFs [4]. The unbalance of this process may result in RHO-GTPase-related cell function deregulations, such as migration, polarity, proliferation and survival, which, in turn, may lead to tumorigenesis.

*RHOA* has long been postulated as an oncogene, mainly due to its ability to transform fibroblasts through induction of Ras [5,6]. The cancer-related role of RHOA has been mostly found in solid tumors, since increased *RHOA* expression was described in hepatocellular, bladder, liver, ovarian, breast, gastric, colon and lung cancer [7]. Additionally, RhoA was reported to be crucial for cell motility, suggesting an important role for this small GTPase in the invasive phenotype of the tumor cells [8,9,10]. However, human cancer genetic studies have shown that the amplification of *RHOA* is rare in human cancers. Instead, recent whole genome sequencing analyses have found recurrent loss-of-function *RHOA* hotspot mutations in gastric cancer [11], angioimmunoblastic T cell lymphoma (AITL) [12,13], adult T-cell leukemia/lymphoma (ATLL) [14], Burkitt lymphoma (BL) [15] and diffuse large B-cell lymphoma (DLBCL) [16], which suggests a rather complex picture of RHOA function depending on cancer cell type.

Another member of the family, *RHOB*, is generally considered to act as a tumor suppressor gene. First of all, its expression is decreased in several tumor types and its presence is inversely correlated with disease progression. Then, the enzyme has been involved in the disruption of several malignant processes, including tumor growth, cell migration and invasion [17]. Conversely, an overexpression of RHOB has been reported in T-acute lymphoblastic leukemia (T-ALL) compared to primary human T cells [18], suggesting that the functions of this GTPase might be context- and tumor-dependent.

Similar to RHOA, RHOC has also been postulated as an oncogene due to its ability to promote invasion and metastasis in several types of cancer through regulation of cell migration and proliferation [19,20]. RHOC is upregulated in T-ALL cells, in which it regulates reactive oxygen species (ROS) production and the subsequent cytoskeleton rearrangement essential for cell migration [21]. Moreover, RHOC is also implicated in the regulation of phagosome formation in macrophages through the modulation of cytoskeletal remodeling via mammalian diaphanous 1 (mDia1) [22].

Given the complex implications related to the role of individual Rho GTPases in several cancer types, this review will focus specifically on RHOA GTPase, emphasizing the recent findings on RHOA signaling in normal and cancer cell biology. We will discuss the implications of RHOA therapeutic targeting, particularly in hematological tumors.

## 2. Physiological Functions of RHOA in the Immune System

Several studies demonstrate that RHOA plays an important part in immune responses and is crucial to the development and activation of immune cells [23]. Figure 1 summarized the different roles of RHOA in the different immune compartments.

Hematopoietic progenitor cells (HPCs) are essential to hematopoiesis and the immune response, and their activation and differentiation are highly regulated and controlled. RHOA has an important role in both hematopoietic stem cells (HSC) and in HPCs [24]. A downregulation of RHOA signaling in vitro by an inhibitor of RHO-associated protein kinase (ROCK), Y-27632 or using small interfering RNA, highlighted some changes involved in the morphology and migration of these stem cells [25,26] and also in the engraftment and self-renewal of the HSCs [27,28]. In vivo experiments upon loss of RHOA activity demonstrated that the GTPase is essential for the multilineage hematopoiesis via regulation of actomyosin signaling, cytokinesis and programmed necrosis [29,30,31].

### 2.1. RHOA GTPase Signaling in Myeloid Cells

In the erythroid lineage, RHOA, other RHO GTPases and CDC42 have been implicated in progenitor commitment and proliferation. RHOA is present in the cytosol and at the membrane of erythrocytes, and its inhibition by Y-27632 results in increased erythrocyte deformability [32,33,34]. At the erythroblast stage, ROCK is found to be activated by caspase-3-mediated cleavage, implying that ROCK is a negative regulator of stress erythropoiesis [35]. Although RAC GTPases promote erythroblast enucleation [35,36], the role of RHOA in this phenomenon is still unclear and an erythroid-specific mouse model will be required to further investigate this issue.

Megakaryocytes are large, polyploid cells of the bone marrow and precursors of platelets. RHOA has been demonstrated to play a critical role in platelet development, as the cytoskeleton rearrangement is important in both the maturation and the initiation of proplatelet formation [37,38]. During cytokinesis, RHOA signaling is required for its completion as it established the actomyosin ring at the cleavage furrow, but it must be downregulated for megakaryocyte polyploidization [39].

In vitro, overexpression of active RHOA resulted in an inhibition of proplatelet formation, suggesting that RHOA is a negative regulator of this process [37,40]. An in vivo deletion of the protein, specifically in megakaryocytes and using the RHOA^fl/fl^PF4CRE^+^ mice model, caused a significant macrothrombocytopenia and the appearance of larger megakaryocytes characterized by a higher mean ploidy and a defective platelet activation [41,42]. These phenomena were due to the contribution of RHOA to the activation of platelet integrins and to the structure of microtubules in spread platelets; however, the exact mechanism(s) by which RhoA contributes to platelet activation is still under debate [41].

In the myeloid lineage, polymononuclear neutrophils (PMN) are innate immune cells rapidly recruited to sites of invading pathogens and local inflammation. Directed migration towards a gradient of chemoattractant is crucial for the neutrophils. While RHOA expression is reduced in the leading edge, activated RHOA promotes myosin cytoskeleton reorganization at the rear and regulates actomyosin contraction [43,44,45,46]. Its downstream effector, ROCK, regulates cell spreading and uropod formation [47]. Francis et al. demonstrated that mice with lower RHOA activity due to the absence of the RHOA GEF, Lsc, neutrophils showed undirected migration due to a deficient pseudopod formation [48], indicating that RHOA is an essential factor for the maintenance of neutrophil polarity and efficient migration [49].

In eosinophils, the RHO–ROCK axis is necessary for migration and uropod detachment by stimulation of actomyosin filament contractions. Together with RAC and CDC42 in the leading edge, the coordination of the three GTPases are necessary for correct cell migration [50].

Conventional dendritic cells (cDCs) are a subtype of dendritic cells, which act as antigen-presenting cells (APCs). Li et al. nicely demonstrated that a loss of *RhoA* specifically within DCs, using the CD11cCRE+ mice model, resulted in a decrease in the number of these cells due to cytokinesis failure and increased apoptosis, identifying *RhoA* as an important regulator in the homeostasis and differentiation of DCs [51]. Additionally, a deficiency in the *RhoA* regulator Arhgef impaired the chemotaxis and migration of immature DCs in vivo [52] and affected both dendrite formation and motility [53]. Since DCs interact with T cells via the immunological synapse, Shurin G. et al. investigated the formation of these latter and found that *RhoA*, along with *Rac1*, were crucial regulators of this process and therefore regulated the DC-mediated response and antigen presentation [54].

Macrophages contribute to tissue homeostasis. Depending on microenvironmental stimuli, monocytes can undergo a polarization towards an M1 macrophage phenotype, characterized by proinflammatory and phagocytic properties, or M2 macrophages, with a more anti-inflammatory phenotype. Liu et al. demonstrated that M0 and M2 macrophages have significantly higher levels of *RHOA*, and that interfering with the RHOA pathway produced a different polarization and morphological adaptation of M0 and M2 macrophages [55,56]. Indeed, *RhoA*^−/−^ macrophages showed a defect in their migration property, which simultaneously required the activation of *RhoA* in the uropod and an inhibition of the GTPase in the podosome to control and direct this migration [57].

During phagocytosis, RHOA is found to be upregulated just prior to the cell engulfment, suggesting that a RAC1/RHOA balance can modulate the critical point of phagocytosis via remodeling of the cytoskeleton [58,59].

### 2.2. RHOA GTPase in the Control of Lymphoid Lineage

During T cell development, a specific deletion of *RhoA* in the T cell lineage leads to a defective thymocyte development and a general decrease in the number of immune cells. This effect can be explained partly by the role of RHOA in the survival and the proliferation of T lymphocytes [60]. Moreover, during immune response and inflammation, T cells need to cross endothelial barriers, thereby triggering cell transmigration and polarization, two processes in which RHOA plays a crucial role [61,62]. Heasman et al. described how RHOA is activated both in the leading and rear edge of the T cell for contraction and retraction and that a loss of RHOA led to a loss of this migratory polarity [63]. As mentioned above, RHOA is important for immune synapse formation. An inactivation of RHOA in Jurkat cells inhibited the CD82-induced changes that are necessary for an activation of the T cell receptor (TCR) signaling on T lymphocytes [64].

Additionally, over the past few years, new atypical negative regulators of RHOGTPases have been identified in immune cells [65]. FAM65B, a member of the FAM65 family proteins, also known as RIPOR (RHO family interacting cell polarization regulator) proteins, has been described to bind specifically to RHOA (but not to CDC42 or RAC1) [65]. FAM65B is associated with a quiescent state of T lymphocytes, as it is expressed at high levels by naive cells, whereas activated T cells exhibit a complete loss of this factor. FAM65B was found to be responsible for RHOA activation in resting T cells and has been involved in the inhibition of T cell proliferation upon antigen recognition [66]. Interestingly, chemokine stimulation leads to FAM65B phosphorylation, thereby impeding the binding activity of this latter to RHOA. This process culminates in RHOA activation, which favors T cell motility through cytoskeletal remodeling [67].

RHOA also plays a crucial role in B cell development. In the earlier stages of the B cell differentiation process, a specific deletion of RHOA in the HSC evokes a reduction in the immature B cell and pro-/pre-B cell populations within the bone marrow. A deletion of *RhoA* specifically in the CD19+ cell population impaired B cell development in the spleen, as shown by a decrease in the percentage of the transitional, follicular and marginal zone B cells [68]. B-cell activating factor receptor (BAFF-R), an important regulator of cell survival, is reduced in *RhoA*^−/−^ splenic cells, supporting an important role of RHOA in B cell proliferation and differentiation [68]. Finally, B cell receptor (BCR) and phosphoinositide 3-kinase (PI3K) signaling have been described to activate RHOA and, therefore, to promote cell proliferation [69], suggesting this Rho GTPase could be an important regulator of other B cell functions.

## 3. RhoA in Hematopoietic Tissues: Oncogene or Tumor Suppressor?

RHOA was initially postulated as a pro-oncogenic signal transducer in epithelial cells [70]. Several studies strengthened this hypothesis, showing that *RHOA* is upregulated in solid tumors, including testicular [71], breast [72,73], colon [73], lung [73] and head and neck squamous cell carcinoma [74]. Moreover, it has been reported that an altered RHOA activity is associated with cancer progression, invasion and angiogenesis in various human cancers. Indeed, *RHOA* knockdown decreased tumor growth and angiogenesis in a breast cancer MDA-MB-231-xenograft mice model [73,75]. Similarly, RHOA over-activation was observed in gastric tissue and cell lines and has a proven role as a regulator of G1 to S cell cycle transition. Mechanistically, it was observed that the knockdown of mDIA1, a well-known RHOA effector, increases the intracellular levels of the cyclin-dependent kinase (CDK) inhibitors, p21Waf1/Cip1 and p27Kip1, leading to cell cycle blockade. On the other hand, the pharmacological depletion of ROCK led to the downregulation of CDK4 and CDK6 at both mRNA and protein levels [76,77]. Conversely, the functional relevance of RHOA was also shown in lung cancer, where its deletion stimulated lung adenoma formation with a more aggressive phenotype. Of special interest, *RhoA* and *RhoC* deletion significantly impaired tumor formation, suggesting that the ablation of the GTPase might lead to a pro-tumoral compensatory mechanism [78].

High-throughput sequencing technologies have identified several gain- and loss-of-function hotspot mutations within the *RHOA* gene, suggesting, once again, that the GTPase may have tumor-specific functions. Somatic *RHOA* mutations have been described in about 25% of diffuse gastric cancer patients, affecting R5, G17, Y42 and L57 residues [11,79,80]. Among them, the most frequent mutation is the oncogenic Y42C mutation, which affects the RHOA-effector interactions [81]. The enzyme is also mutated in lung and bladder cancer, with E47K being the mutational hotspot (www.cbioportal.org (accessed on 1 November 2022)). Moreover, *RHOA* mutation and deletions are also observed in breast cancer, however, at low frequency. It has been suggested that E40Q could be a mutational hotspot, mainly because the same alteration has been frequently described in head-and-neck squamous cell cancer [82]. Although these studies show that RHOA might be implicated in the malignant transformation of solid tumors, recent evidence has demonstrated its tumor suppressor activity in some hematological malignancies.

The development of hematological cancers is a multistep process that consists of dysregulation of cell proliferation, cell death inhibition, cell immortalization, invasion, metastasis and the accumulation of mutations in key genes that may cause primary tumor formation. In 2014, Palomero et al. [83] reported a recurrent mutation in RHOA (G17V) present in 67% of AITL and 18% of peripheral T cell lymphoma (PTCL). The glycine at the 17th position mutation disrupts RHOA interaction with GEFs due to loss of nucleotide binding, which leads to the loss of GTPase activity and inhibition of stress fiber formation (Figure 2). Subsequently, two other groups [13,84] corroborated these findings, showing that RHOA-G17V is a loss-of-function mutation able to drive AITL with high proliferative and invasive capacity. Consistently, Cortes et al. [85] have shown that RHOA-G17V mutation drives proliferation and polarization of human follicular helper T lymphocytes (Tfh), which eventually culminate in AITL development with the upregulation of ICOS (inducible co-stimulator) and activation of PI3K/mitogen-activated protein kinase (MAPK) signaling pathways.

Vav guanine nucleotide exchange factor (VAV1) is a protein stimulated by the engagement of the TCR through GEF activity and is also reported to be mutated in human AITL [86,87]. Interestingly, it has been shown that RHOA-G17V specifically binds to the DH domain of VAV1, thus modulating its function as an adaptor to regulate the TCR signaling complex in AITL [88]. Furthermore, dasatinib, a multikinase inhibitor used to treat some types of leukemias, efficiently blocks VAV1 activation and subsequent RHOA-G17V signaling [88], pointing out a new possible therapeutic approach for AITL patients.

Another potential inactivating RHOA mutation, R5Q, has also been reported in a cohort of pediatric BL patients [15]. Recent evidence based on functional assays showed that, indeed, RHOA-R5Q mutation impairs the activity of the GTPase in BL and DLBCL [16]. Additionally, based on deep sequencing of 203 ATLL samples, Nagata Y et al. have shown that ~15% of cases also harbor recurrent *RHOA* mutations in the GTP binding domain with different or even opposite functional consequences in terms of RHOA activity, suggesting that both loss- and gain-of-function mutations might be involved in leukemogenesis [14]. Moreover, the data published on the Cancer Genome Atlas database (TCGA; www.cancergenome.nih.gov (accessed on 1 November 2022); www.cbioportal.org (accessed on 1 November 2022)) show that the *RHOA* gene is rarely amplified and is often deleted in different cancer types, and that *RHOA* mutations are associated with a worse overall survival of non-Hodgkin lymphoma (NHL) patients. On the other hand, it has been shown that Wnt family member 16 (WNT16b) activates RHOA signaling via a non-canonical WNT pathway and that WNT5a-receptor tyrosine kinase like orphan receptor 1 (ROR1) signaling enhances the proliferation of B-cell precursor acute lymphoblastic leukemia (BCP-ALL) cells, via RHOA and signal transducer and activator of transcription 3 (STAT3) upregulation [89]. Finally, blockage of RHOA activity has been involved in selenite-induced apoptosis of leukemia cells in both in vitro and in vivo models, highlighting the role of the RHOA pathway in the survival of hematologic malignant cells [90].

*Clostridium botulinum*-derived toxin C3 transferase has long been used to inhibit RHOA activity by abolishing its interaction with downstream effectors. Interestingly, Cleverley S et al. [91] used transgenic mice with thymocyte-specific expression of the enzyme C3-transferase to demonstrate that the loss of RHO function in the thymus resulted in a partial blockade of pre-T cell cycle progression and, eventually, in the development of aggressive thymic lymphoblastic lymphomas. This study suggests that RHOA may act as a suppressor of lymphoid cell transformation. Importantly, mice deficient specifically in regulatory T cells (Treg)-associated *RhoA* displayed lymphadenopathy, high leukocyte infiltration and an increase in effector T cells and effector memory T cells. *RhoA* deletion in these mice resulted in systemic autoimmune responses and fatal systemic inflammatory disorders [92].

It was recently shown that miR-126-3p and miR-145-5p can repress RHOA activity in AITL through the regulation of ROCK and S1PR2, a RHOA upstream receptor. Importantly, high levels of miR-126-3p and miR-145-5p were associated with a poor clinical outcome of AITL patients [93].

A recent study showed that CD47, a cell surface transmembrane protein that inhibits phagocytosis, is associated with T-cell lymphoma tumor formation and metastasis by upregulating A-kinase anchor protein 13 (AKAP13)-mediated RHOA activation, supporting the notion that RHOA may exert an oncogenic function in this malignancy [94]. Additionally, Pan Y et al. [95] demonstrated the crucial role of RHOA activation for DLBCL amoeboid motility through activating STAT3, which is a critical hallmark of late-stage progressing disease.

Thus, the recurrent and dominant negative nature of *RHOA* mutations in AITL, PTCL, BL and DLBCL strongly support a tumor suppressive role for RHOA in hematological cancers. However, the current knowledge on the role of wild-type *RHOA* in these entities is conflicting and requires additional clarifying evidence in order to obtain a strong rationale for using RHOA-targeting therapy in these tumors.

## 4. Bioinformatic Prediction of RHOA Therapeutic Agents

The use of bioinformatics tools takes advantage of the abundant amount of available data obtained by different experimental techniques and the power of computational processing [96], reducing time and costs and increasing accuracy. In silico methods for therapeutic target discovery are based on genomic comparative analysis and/or network-based methods. Although the Rho family of GTPases has been linked to different cancers using bioinformatics methods [97,98], to date there are not many applications centered on the role and/or activity of RHOA in hematological cancers.

In comparative genomics [99], the DNA sequence or structure of the phenotypes of interest are used [100]. Whether it is the target gene or a gene that indirectly regulates it, these methods allow us to identify conserved sequences or specific sequences for each phenotype associated with new therapeutic targets or resistance/toxicity. In addition to a direct comparison between conditions of interest, the extensive information deposited in databases (i.e., genetic, metabolic and proteomic) can now be used to address and interpret information from new studies. For this, first, the affected pathway(s) must be defined, using databases such as KEGG [101], and the proteins/genes with differential activity between the phenotypes studied must be selected according to the objectives of the study; then, the information (i.e., sequences) of the important genes or proteins must be retrieved (i.e., using the UniProt database [102]) and some local alignment tool (i.e., BLASTp for proteins and BLASTn for nucleotides) is used to localize and quantify the differences in the compared sequences. Finally, it is necessary to evaluate and validate if the variants (i.e., SNPs) are relevant for patient stratification or targetable.

Network-based methods [103,104] provide two main advantages: integration of different types of data (i.e., clinical, omic, pharmacological) and structuring of information where elements, such as genes or proteins, are represented by nodes, and the relationship between them, qualitative or quantitative, are represented by edges. The focus of this type of analysis can be by centrality, where nodes with more edges and/or are more central in the network would be essential elements for such a network, and by dissimilarity, where elements of the network are related differently or lose relationship, indicating different states of the network for different conditions.

A large number of resources such as databases and bioinformatics tools are available to implement the previously mentioned models. UniProt [102], DEG [105] and RefSeq [106] contain sequence and annotation information for proteins, DNA or both, respectively. Platforms such as KEGG [101] and Reactome [107] contain descriptions of genes and related biomolecules represented as networks of pathways elaborated under some biological criteria (i.e., cell migration, inflammatory response). Other databases are specialized in interactions, such as STRING [108] and MINT [108] for proteins, IntAct [109] and BioGrid [110] that consider other biomolecules such as nucleotides, or miRTarBar [111] for miRNA and their targets. Other databases, such as GEO [112], DrugBank [113], GO [114] and DAVID [115] contain detailed information on molecules of interest that can help in data interpretation. Regarding data processing tools, BLAST [116] and its specialized versions offer a set of very useful analyses and results when comparing amino acid or nucleotide sequences and their functional and evolutionary relationship; MUSCLE [117] is a program that can be used to perform multiple nucleotide or protein alignments with high performance. For network-based methods, we use programs that facilitate visualization and provide structural analysis and enrichment tools, such as Cytoscape [118] and Gephi [119], or platforms that allow us to perform the same tasks with the additional feature of having direct links to databases that facilitate our exploratory analysis, such as NetworkAnalyst [120], Pathwaylinker [121], BioCyc [122], among others.

Some recent studies have applied these methods to the study of RHOA. For example, in one study, different analytical techniques were used to demonstrate variations in genomic sequence (SNV and indels) and structure (CNV), with biochemical validations and docking simulation of the loss of activity to finally integrate these data with gene expression levels and propose a characteristic pathway of AITL caused by the G17V mutation in RHOA [13]. In another study [123], consensus gene sets were defined using the gene set enrichment analysis (GSEA) approach in three collections of gene sets (KEEG, hallmarks and GO) to evidence other regulatory functions of the GTPase, such as cell growth, vesicular transport and DNA replication. Using a network approach and public data mining, Fathima and collaborators defined a list of proteins as potential anticancer drug targets, among which RHOA was identified [124]. For this aim, the study extracted a list of genes relevant in non-apoptotic cell death from literature mining and transcriptome data of some cancers deposited in NCBI’s Gene Expression (GEO, https://www.ncbi.nlm.nih.gov/geo/ (accessed on 1 July 2022)). Then, this list was used as input data for transcription factor (TF)–target and protein–protein interaction databases to then integrate the information in networks and quantify the topological relevance of each element. The most important proteins were queried in DrugBank (https://go.drugbank.com/ (accessed on 1 July 2022)) to find the list of related drugs and proteins that can be considered as new therapeutic targets. Furthermore, miRNAs that could regulate drug target genes were queried. Finally, relevant compounds were used for pathway enrichment and for structural simulation of possible drug binding sites. These types of studies not only evidenced and quantified the importance of proteins such as RHOA in cancer, but also enabled the depicting of their relationship with other biomolecules, structuring abundant information currently available and helping to plan future in vitro/in vivo studies.

## 5. Approaches to Modulate RHOA Signaling in Hematological Cancers

As illustrated above, the numerous pathways regulated by the RHO GTPase family define multiple aspects important for the proliferation and propagation of malignant cells. Therefore, both RHO GTPases themselves and downstream effectors are attractive targets for therapeutic drug design in cancer (Figure 2).

First, and since RHO GTPases are considered undruggable as they lack stable cavities beside the nucleotide binding pockets, several compounds have been developed, with the ability to modulate multiple stages of the GDP/GTP cycle [125,126]. As stated before, the bacterial exoenzyme C3 is a transferase extracted from Clostridium botulinum that inhibits RHOA, RHOB and RHOC by adding an ADP-ribose moiety on asparagine residues, thereby blocking the small GTPases in an inactive state and keeping them bound to GDI [127]. Since the cell penetrability of the wild-type C3 transferase was very low, and that this factor introduced covalent modifications that limited its clinical development, the cell-permeable C3 transferase Cethrin (BA-210/VX-210) was further developed. Given the importance of the Rho signaling pathway in growth cones, this compound was first shown to prevent damage after spinal cord injury [128]. In vitro studies further demonstrated that blockade of RHOA with Cethrin could enhance the adhesion of T-ALL cells in vitro, a phenomenon that required the polymerization of actin filaments, thus suggesting that activation of RHOA kinase could promote leukocyte deadhesion by inhibiting integrin- and cytoskeletal-dependent spreading of malignant leukemic cells [129].

Other compounds showed activity in preclinical models of hematological neoplasms, such as the inhibitors of RHO GTPase-GEF binding. EHop-016 is an antagonist of RAC1–VAVv2 interaction [130] that can block GTPase activity, thereby improving the antitumor effect of cisplatin in mouse models of esophageal squamous cell carcinoma [131,132]. Validating potent effects of RAC1 inhibitors in acute myeloid leukemia (AML), Hemsing and collaborators recently reported that primary AML cultures, and especially cases harboring a nucleophosmin 1 (*NPM1*) mutation, presented a good responsiveness to EHop-016 and to its precursor, NSC23766, as well as to three additional RAC1 inhibitors (ZINC69391, ITX3 and 1A-116), with IC_50_ in the low micromolar range of concentrations [133]. While EHop-016 was developed as an improved NSC23766-derived RAC1 antagonist [134], the parental compound was initially aimed at blocking a surface groove that is critical for the interaction of RAC1 and its associated GEF, Tiam1, without affecting the activity of CDC42 or RHOA [135]. In a preclinical model of chronic myeloid leukemia (CML), either sensitive or resistant to the tyrosine kinase inhibitor (TKI) imatinib, directed against the BCR-ABL fusion gene, NSC23766 displayed an antiproliferative effect in vitro and in vivo and blocked malignant cell dissemination in xenograft studies, supporting the notion that during CML pathogenesis, RHO GTPases can be activated by p210-BCR-ABL [136]. Based on the structure of NSC23766, additional blockers of RAC1 and CDC42 activation have been designed in addition to EHop-016, such as AZA1, which inhibits cellular migration of prostate cancer cells in vivo [137]. Similarly, the CDC42-targeting agent AZA197 impaired the motility of colorectal carcinoma cells [138]. In parallel, the CDC42 activity-specific small molecule inhibitors, Casin and ZCL278, specifically blocked the CDC42 interplay with its GEF, intersectin, suppressing actin-based motility and cell migration without affecting other RHO GTPase activity [139,140]. While ZCL278 was remarkably active in slowing the development of lung cancer in mice [141], the treatment of aged HSCs in a transplantation model allowed for the reestablishment of an immune system similar to that of young animals [142]. In parallel, non-competitive inhibitors, ML141 and EHT1864, targeting, respectively, CDC42 and RAC1, are able to dissociate the bound nucleotide, thus inactivating these GTPases [143]. Of note, the allosteric inhibition of CDC42 signal by the highly selective ML141 [144] revealed that the GTPase is a negative regulator of granulocyte colony-stimulating factor (G-CSF)-mediated HSC mobilization and that the inhibition of its activity by this compound can improve mobilization efficiency [145]. Like EHop-016, EHT1864 interferes with the migration of metastatic breast cancer cells [130,146]. In vitro exposure of AML cells to this agent triggered a time- and dose-dependent cell cycle arrest at the G1 phase, followed by autophagic cell death [147]. These changes were concomitant to alterations in p53 and cyclins expression levels, and to the disruption of the PI3K signaling axis. Interestingly, the combined treatment of EHT1864 and low dose daunorubicin enhanced AML cell apoptosis.

Among the best strategies to prevent the binding of RHOA with its GEFs and to suppress its GTPase activity, the leukemia-associated RHOA GEF, LARG, has been the target of a virtual screening using published protein:protein interactions. This screening identified Rhosin as a compound with the capacity to inhibit specifically the GEF-catalyzed RHOA activation, although it was not selective for RHOA-LARG, as it could also affect other RHO GEFs, such as Dbl, p115RHO GEF and PDZ-RHO GEF [148]. While Rhosin demonstrated interesting preclinical activity in chemotherapy-resistant gastric tumors, in breast cancer and in melanoma [149,150], its suitability for the treatment of blood cancers remains unknown. Y16 is another compound identified through high-throughput screening and is characterized by the capacity to inhibit RGS-containing RHO GEFs. In breast cancer models, this compound could block RHOA activity and prevent the formation of mammary spheroids [151,152].

RHO GTPase activity is tightly dependent on the anchorage of these enzymes at the plasma membrane, with this process being regulated by post-translational modifications that include isoprenylation and palmitoylation [153]. In this context, a specific and first-in-class inhibitor targeting geranylgeranyl transferase-1 (GGT-1) activity, PTX-100, is considered the only RHOA inhibitor in clinical development. Indeed, having demonstrated significant antitumor activity in mouse models of breast cancer [154], this agent was well-tolerated and achieved stable disease in a phase 1 trial in treatment-refractory advanced solid tumors [155]. Currently, PTX-100 is included in a pharmacodynamic and pharmacokinetic basket study involving cancer patients with advanced solid and hematological malignancies, including cases with relapsed/refractory PTCL or multiple myeloma (MM) (NCT03900442). The completion of this study is planned for April 2023.

In addition to this approach, RHO GTPase membrane localization can also be targeted indirectly with statins, as these molecules alter cholesterol biosynthesis by inhibiting 3-hydroxy-3-methylglutaryl-coenzyme A reductase (HMGCR), thereby reducing the intracellular levels of farnesyl pyrophosphate and geranylgeranyl pyrophosphate and retaining the RHO GTPases within the cytosol [156,157]. Statins have been used for years in the treatment of patients with cardiovascular affectations [158]. However, these agents may also target different processes involved in tumor development, such as cancer cell growth, migration, inflammation and angiogenesis, promoting, at the same time, the apoptotic death of the neoplastic cells and reducing cancer-associated mortality [159]. Therefore, this class of drugs plays a positive role in the chemoprevention of cancer and represents excellent candidates for drug repurposing in the management of patients with gynecological [160], brain [161], gastrointestinal [162] or prostate cancer [163]. In hematological neoplasms, a recent meta-analysis of 14 clinical studies did not support a potential role of statins in the prevention of cancer development, with no evidence of an association between statin use and the frequency of hematological disorders in either randomized clinical trials or observational studies [164]. Nonetheless, several preclinical studies have demonstrated the pleiotropic effects of these drugs, as they can elicit apoptosis, block angiogenesis, exert immunomodulatory effects and overcome chemoresistance in different subtypes of leukemias. Indeed, in preclinical models of CML, statins were found to exert an anti-leukemic effect and to synergize with BCR-ABL targeting agents [165,166]. In chronic lymphocytic leukemia (CLL), although some studies suggested that statins may interfere with anti-CD20 therapy due to their ability to alter the conformation of the membrane receptor [167], both lovastatin and simvastatin have been shown to activate the intrinsic apoptotic pathway [168] and to impair mitogen-induced proliferation of primary cultures [169]. However, these promising preclinical results were not validated in clinical settings. Supporting a lack of activity of statins on the natural CLL clinical course, neither time to first treatment of newly diagnosed CLL patients nor the efficacy of anti-CD20 therapy, overall survival or treatment-free survival were dependent on whether or not patients were receiving a statin at diagnosis [170,171]. Among the B-cell subtypes of NHL, and in particular, in DLBCL, which accounts for the most prevalent B-NHL entity, the added values of statins are controversial. Indeed, while patients treated with high-dose statins associated with a standard immunochemotherapeutic (R-CHOP) regimen experienced a significantly higher complete response (CR) rate when compared to patients receiving R-CHOP monotherapy, with an improved 6-years progression free survival (PFS) [172], its seems that this assumption may be ethnicity-dependent. Indeed, in a recent meta-analysis collecting the data from almost 10.000 DLBCL patients, a subgroup analysis failed to show that statin use was associated with overall survival (OS) of patients from Asia, whereas statins were found to be associated with a significantly improved OS of patients from Western countries [173]. Very recently, it has been described that simvastatin exerts single agent activity in in vitro and in vivo models of mantle cell lymphoma (MCL), a rare but very aggressive subtype of B-NHL. In this study, simvastatin exhibited a pleiotropic effect against several MCL hallmarks, including deregulated cell cycle progression, apoptosis, migration and invasion, warranting further clinical evaluation of this agent in clinical settings [174]. Finally, AML is probably the entity that may benefit the most from statin-based therapies. In preclinical models of the disease, statins have been shown to exert single agent activity and to improve the activity of conventional cytotoxic therapy through different mechanisms that involved the modulation of isoprenylation, cholesterol metabolism, leukemic cell differentiation, apoptotic signaling, regulation of protein kinase C (PKC) signaling, intracellular accumulation of intracellular Fms-related receptor tyrosine kinase 3-internal tandem duplication (FLT3-ITD) and consequent blockade of MAPK and AKT signaling [175]. In a phase 1 study assessing the combination of pravastatin with idarubicin-high dose AraC (Ida-HIDAC) including a significant percentage of AML patients with unfavorable cytogenetic/molecular features and/or in salvage therapy, the maximum tolerated dose (MTD) was not reached and the response rate was 30%, while approximately 25% of patients were able to proceed to allogeneic HSC transplant [176]. These encouraging results led to a phase 2 tevaluation of this combination, which demonstrated an impressive rate of CR and CR with incomplete count recovery (CRi) of 75% in this subgroup of patients [177].

Considering the possibility to target RHO GTPase downstream signaling, and in particular the cancer cell metastatic properties, ROCK has emerged as the most studied downstream effector. The antitumor activity of the ROCK inhibitors Y27632 and fasudil have been well characterized and studied preclinically in different types of solid tumors [178,179]. Fasudil exhibits antagonistic activity against serine–threonine kinases, including ROCK [180], to which it binds through the ATP-binding pocket. Fasudil is, so far, the only approved RHO GTPase inhibitor, although in a cancer-unrelated indication (i.e., for the prevention of vasospasm after subarachnoid hemorrhage in Japan). This clinically safe drug demonstrated the capacity to partially control white cell and monocyte counts and to prolong the survival of some animals in a mouse model of Cbl/Cbl-b double knockout (DKO)-associated myeloproliferative disorder [181]. Y-27632 activity on the regulation of the cytoskeleton was first demonstrated in inflammatory diseases [182] and then exploited later on to mitigate WNT5A-dependent cell migration in preclinical models of classical Hodgkin’s disease (cHL) [183]. More selective inhibitors of ROCK include OXA-06 and PT262, RKI-1447 CCT129253 and AT13148, but their evaluation as anti-metastatic agents has been restricted so far to non-hematological diseases [184,185,186,187]. Of note, due to controversial reports on the possible pro-tumoral activity of some of these agents, the sole phase 1 dose-escalation trial carried out in the last years (NCT01585701) is evaluating AT13148 for the treatment of cancer patients with advanced disease. At the maximum dose of 240mg, the most common drug-related toxicities in this study so far have been nausea, anorexia, headache and hypotension. At the present time, data collection has been completed and patient outcome is under evaluation.

Another downstream effector that has been extensively studied to counteract RHO GTPase signaling is p21 (RAC1) activated kinase 1 (PAK1) [188]. IPA-3 is a non-competitive antagonist designed to target PAKs independently of ATP by adding a sulfhydryl moiety to the N-terminal region, thereby promoting the inhibitory conformation [189]. Using a primary culture of progenitor cells and normal hematopoietic counterparts isolated from the bone marrow of newly diagnosed CML patients in chronic phase and CML-blast phase cell lines, IPA-3-mediated targeting of PAK1 and/or PAK2 enhanced the antiproliferative and pro-apoptotic effect of imatinib, without affecting normal counterparts [190]. Due to the unfavorable pharmacological properties of IPA-3, several allosteric inhibitors of PAKs have been designed [191]. Among these molecules, the ATP-competitive PF-3758309 showed promising results in different in vitro and in vivo cancer models, especially in murine and human models of AML and myeloproliferative neoplasms [192]. In this study, the authors demonstrated that the focal adhesion kinase (FAK)/RAC1-TIAM1/PAK1 axis plays a crucial role in the transformation induced by FLT3-ITD and the oncogenic form of KIT (KIT^D816V^), and that targeting PAK1 in oncogene-bearing cells in vitro or in vivo inhibits the presence of active STAT5 in the nuclear compartment. This phenomenon delayed the onset of leukemia by repressing the expression of STAT5-responsive genes. PF-3758309 entered into clinical evaluation in 2009, but the study was closed prematurely due to pharmacokinetic issues [193]. A second molecule, called FRAX597, identified by high-throughput screening and showing potent antitumor activity in preclinical models of Ras-dependent cancers [194], also exhibited potent leukemia inhibitory effects in in vitro and in vivo models of AML, affecting both the leukemic blasts and the more immature, leukemic stem cell (LSC)-enriched populations. The effect of the drug was mediated through the induction of AML cell differentiation and apoptosis, and the modulation of the MYC transcriptional network [195].

## 6. Conclusions

The recent deciphering of activating or inactivating mutations affecting the RHOA GTPase and/or its complexes networks in hematological cancers strongly support the notion that, similarly to solid tumors, RHOA can exert pro- or anti-tumorigenic functions depending on the cell context and tumor type. In this sense, in determined B-NHL subtypes rather than in chronic or acute leukemias, the enzyme may prevent the development and dissemination of the malignant cells. In order to unravel the identity of key therapeutic targets involved in aberrant RHOA signaling, it will be necessary to achieve a better understanding of the regulating proteins and of the mechanisms underlying the selective recruitment of RHOA downstream effectors in individual disease subtypes. Since a few ATP competitive and non-competitive inhibitors have been developed and entered into the clinic following standard drug development procedures, the implementation of new bioinformatics tools such as networks-based methods and multiomic analyses may be crucial for the design of improved drug candidates that could reasonably be tested in determined hematological diseases.

## Figures and Tables

**Figure 1 cells-12-00433-f001:**
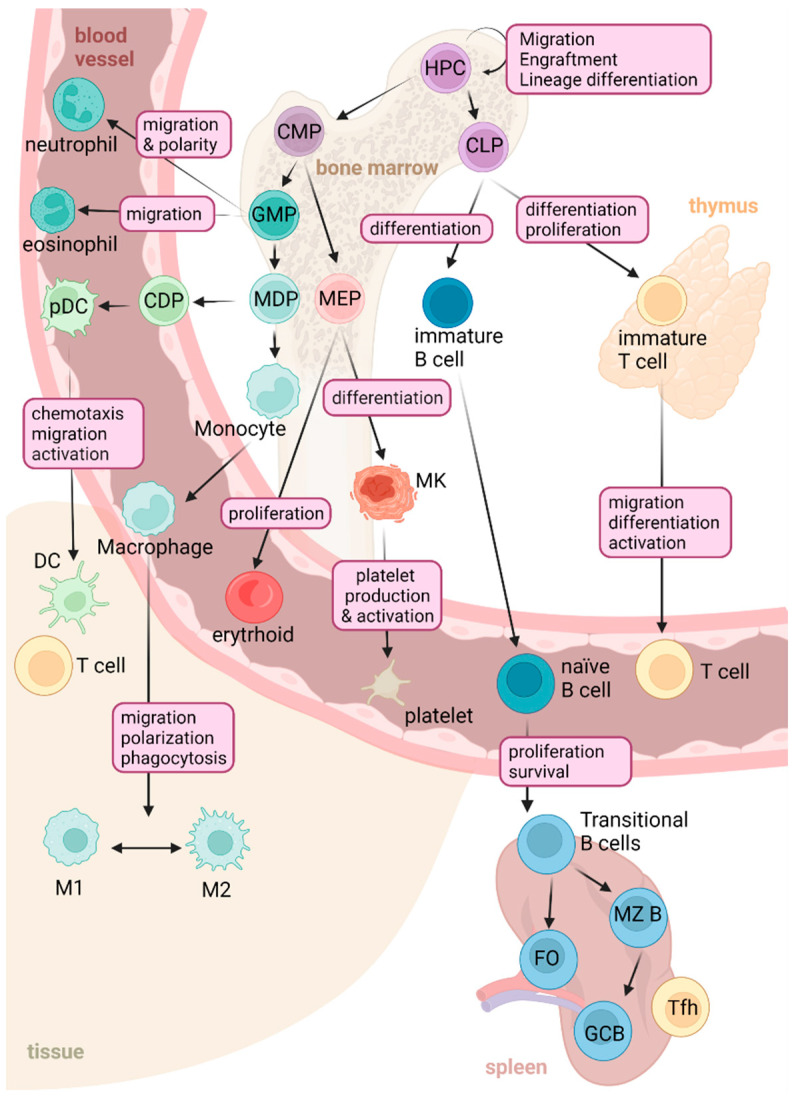
Functions of RhoA signaling in the immune system.

**Figure 2 cells-12-00433-f002:**
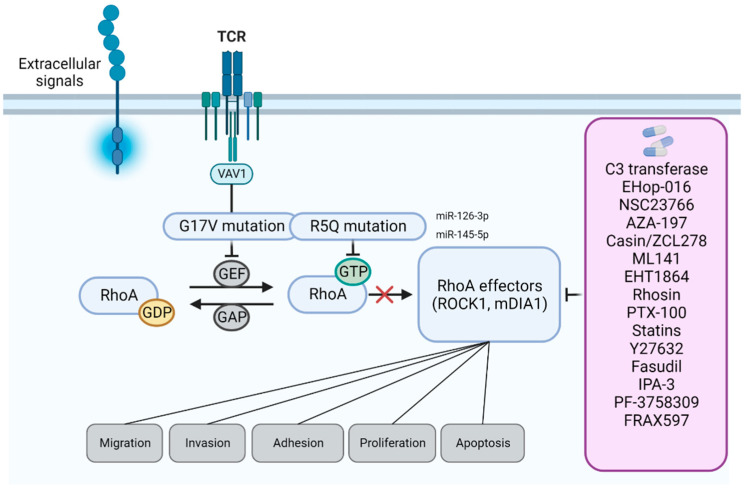
RhoA alterations and therapeutic targeting RhoA-associated signaling in hematological cancers.

## Data Availability

Not applicable.

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
