# Peer review of "RHOA Therapeutic Targeting in Hematological Cancers"

_cells, 2023, doi:10.3390/cells12030433_

Round 1

Reviewer 1 Report

This is a comprehensive and well-written review on RhoA GTPase as a therapeutic target in hematological malignancies.

The included figure is of great usefulness.

I would suggest adding a paragraph about the other members of the RhoA family of GTPases (RhoB and RhoC).

Minors:

Page 7 line 265. Correct the following sentence: [...] RhoA may exert have an oncogenic function [...]. "have" should be eliminated.

Author Response

Comment 1) I would suggest adding a paragraph about the other members of the RhoA family of GTPases (RhoB and RhoC).

Answer)  Following reviewer’s recommendation, we have included within the introduction a chapter detailing the roles of RhoB and RhoC in cancer (lines 59-72 of the present revised manuscript:

“Another member of the family, RhoB, is generally considered to act as a tumor suppressor gene. First of all, its expression is decreased in several tumor types and its presence is inversely correlated with disease progression. Then, the enzyme has been involved in the disruption of several malignant processes, including tumor growth, cell migration and invasion[17]. Conversely, an overexpression of RhoB has been reported in T-acute lymphoblastic leukemia (T-ALL) compared to primary human T cells [18], suggesting that the functions of this GTPase might be context- and tumor-dependent.

Similar to RhoA, RhoC has also been postulated as an oncogene due to its ability to promote invasion and metastasis in several types of cancer through regulation of cell migration and proliferation [19,20]. RhoC is upregulated in T-ALL cells, in which it regulates reactive oxygen species (ROS) production and the subsequent cytoskeleton rearrangement essential for cell migration [21]. Moreover, RhoC is also implicated in the regulation of phagosome formation in macrophages through the modulation of cytoskeletal remodeling via mammalian diaphanous 1 (mDia1) [22].”

Comment 2)  Page 7 line 265. Correct the following sentence: [...] RhoA may exert have an oncogenic function [...]. "have" should be eliminated.

Answer) we are grateful to the reviewer for pointing out tis typo. The text has been corrected accordingly (line 293 of the present revised manuscript).

Reviewer 2 Report

This review by Santos et al. is clear, well written and most of all well organized. The schematic picture is very nice, may the authors add another one ? Indeed for a review, 2 schemes are usually done and not only one.

Also, while cited references are quite clear and relevant to this work, 2 are missing and should definitely be added: 

  • - 10.1038/s41577-021-00500-7

10.3389/fimmu.2018.02001

Author Response

Comment 1) The schematic picture is very nice, may the authors add another one ? Indeed for a review, 2 schemes are usually done and not only one.

Answer)  We grateful thank the reviewer for this suggestion. On his/her request, we have included a second figure titled “RhoA alterations and therapeutic targeting RhoA-associated signaling in hematological cancers” to illustrate the different approaches undertaken to tackle RhoA oncogenic activities in leukemia and lymphoma, and embedded on page 7 of the revised manuscript:

Comment 2) Also, while cited references are quite clear and relevant to this work, 2 are missing and should definitely be added: 10.1038/s41577-021-00500-7; 10.3389/fimmu.2018.02001

Answer) To fulfill referee’s query, we have added a new chapter in the present version of the manuscript (lines 178-188), that includes the two suggested references, complemented with a third one focused on the role of p-FAM65B on the regulation of RhoA activity in T cells:

“Additionally, over the past few years new atypical negative regulators of RHO GTPases have been identified in immune cells [65]. FAM65B, a member of the FAM65 family proteins, also known as RIPOR (RHO family interacting cell polarization regulator) proteins, has been described to bind specifically to RhoA (but not to Cdc42 or Rac1) [65]. FAM65B is associated with a quiescent state of T lymphocytes, as it is expressed at high levels by naive cells, whereas activated T cells exhibit a complete loss of this factor. FAM65B was found responsible for RhoA activation in resting T cells and has been involved in the inhibition of T cell proliferation upon antigen recognition [66]. Interestingly, chemokine stimulation leads to FAM65B phosphorylation, thereby impeding the binding activity of this latter to RhoA. This process culminates in RhoA activation, which favors T cell motility through cytoskeletal remodeling [67].”

  1. El Masri, R.; Delon, J. RHO GTPases: from new partners to complex immune syndromes. Nat. Rev. Immunol. 2021, 21, 499–513, doi:10.1038/s41577-021-00500-7.
  2. Froehlich, J.; Versapuech, M.; Megrelis, L.; Largeteau, Q.; Meunier, S.; Tanchot, C.; Bismuth, G.; Delon, J.; Mangeney, M. FAM65B controls the proliferation of transformed and primary T cells. Oncotarget 2016, 7, doi:10.18632/oncotarget.11438.
  3. Megrelis, L.; El Ghoul, E.; Moalli, F.; Versapuech, M.; Cassim, S.; Ruef, N.; Stein, J. V.; Mangeney, M.; Delon, J. Fam65b phosphorylation relieves tonic RhoA inhibition during T cell migration. Front. Immunol. 2018, 9, doi:10.3389/fimmu.2018.02001.
